# A rapid antibody screening haemagglutination test for predicting immunity to SARS-CoV-2 variants of concern

Nina Urke Ertesvåg [1,22], Julie Xiao[2,22], Fan Zhou [1], Sonja Ljostveit[1,3], Helene Sandnes[3], Sarah Lartey[1,4], Marianne Sævik [5], Lena Hansen[1], Anders Madsen[1,4], Kristin G. I. Mohn [1,5], Elisabeth Fjelltveit[1,4], Jan Stefan Olofsson [1], Tiong Kit Tan [2], Pramila Rijal [2], Lisa Schimanski[2], Siri Øyen[6], Karl Albert Brokstad[7,8], Susanna Dunachie [9], Anni Jämsén[10], William S. James [11], Adam C. Harding[11], Heli Harvala[12], Dung Nguyen[13], David Roberts[14], PHE Virology group*, Maria Zambon [15], Oxford collaborative group*, Alain Townsend[2,22,23✉], Bergen COVID-19 Research Group*, Nina Langeland [3,5,16,22,23✉] & Rebecca Jane Cox [1,3,4,22,23✉]

## Abstract

**Background** Evaluation of susceptibility to emerging SARS-CoV-2 variants of concern (VOC) requires rapid screening tests for neutralising antibodies which provide protection.

**Methods** Firstly, we developed a receptor-binding domain-specific haemagglutination test (HAT) to Wuhan and VOC (alpha, beta, gamma and delta) and compared to pseudotype, microneutralisation and virus neutralisation assays in 835 convalescent sera. Secondly, we investigated the antibody response using the HAT after two doses of mRNA (BNT162b2) vaccination. Sera were collected at baseline, three weeks after the first and second vaccinations from older (80–99 years, $n = 89$) and younger adults (23–77 years, $n = 310$) and compared to convalescent sera from naturally infected individuals (1–89 years, $n = 307$).

**Results** Here we show that HAT antibodies highly correlated with neutralising antibodies ($R = 0.72$–$0.88$) in convalescent sera. Home-dwelling older individuals have significantly lower antibodies to the Wuhan strain after one and two doses of BNT162b2 vaccine than younger adult vaccinees and naturally infected individuals. Moreover, a second vaccine dose boosts and broadens the antibody repertoire to VOC in naïve, not previously infected older and younger adults. Most (72–76%) older adults respond after two vaccinations to alpha and delta, but only 58–62% to beta and gamma, compared to 96–97% of younger vaccinees and 68–76% of infected individuals. Previously infected older individuals have, similarly to younger adults, high antibody titres after one vaccination.

**Conclusions** Overall, HAT provides a surrogate marker for neutralising antibodies, which can be used as a simple inexpensive, rapid test. HAT can be rapidly adaptable to emerging VOC for large-scale evaluation of potentially decreasing vaccine effectiveness.

## Plain language summary

The aim of this study was to rapidly investigate the immune responses after SARS-CoV-2 infection and vaccination of younger adults and the elderly. Antibodies are proteins produced by the immune system that are released into the bloodstream and help fight infections. A simple method using red blood cells obtained from blood was developed and used to detect antibodies to SARS-CoV-2. This test was able to measure protective antibodies to several variants of concern. The elderly had lower antibody responses after vaccination. Two vaccinations induced a broader antibody response to viral variants, similar to the response induced following Covid-19. This antibody detection method could be used as a finger prick test to rapidly detect specific antibodies to emerging variants and enable quick identification of individuals who might benefit from a booster vaccination.

---

A full list of author affiliations appears at the end of the paper.

There is increasing evidence that neutralising antibodies to the receptor binding domain (RBD) on the severe acute respiratory syndrome coronavirus-2 virus (SARS-CoV-2) spike protein represent an immunological correlate of protection[1]. SARS-CoV-2 evolution has been rapid with the ancestral virus and emerging variants of concern (VOC) straining global health care systems. These VOC (alpha (B.1.1.7)[2], beta (B.1.351)[3], gamma (P.1)[4], delta (B.1.617.2)[5] and the recent omicron (B.1.1.529)[6,7] show increased transmissibility, can escape pre-existing immunity and reduce vaccine effectiveness[8–10], with breakthrough infections reported in COVID-19 vaccinees with low neutralising antibodies[11]. There is a need for a rapid low-cost surrogate neutralisation assay, which can be used at a low biosafety level. This assay could be used for large-scale screening to identify vaccinees potentially susceptible to emerging VOC and who would benefit from a booster vaccine dose.

The neutralisation assay with live native virus is the gold standard for evaluating antibodies to VOC[1]. However, neutralising assays are difficult to standardise across laboratories, are time consuming, expensive and require high containment. Therefore, antibody binding and pseudotype virus assays are widely used to study antibody responses[12–14], but still require specialised laboratory facilities.

Here, we correlate the low-cost rapid hemagglutination test (HAT)[15] with neutralisation of the ancestral Wuhan-like strain in two large independent cohorts of infected patients. Further, we confirm the correlation between HAT and neutralising antibodies to VOC. In the HAT assay, the RBD domain is linked to a monomeric anti-erythrocyte single domain nanobody. When polyclonal serum antibodies bind to the RBD they cross-link and agglutinate the erythrocytes, which can be read visually after one hour. The HAT has a specificity of >99% for detection of convalescent antibodies after polymerase chain reaction (PCR) confirmed infection[15,16]. For influenza, a correlate of protection (COP) has been defined as a haemagglutination inhibition (HAI) titre of 40 for 50% protection from infection. If a similar COP could be established for HAT, it would allow simple standardised evaluation of susceptibility to SARS-CoV-2 infection and waning vaccine responses to VOC to guide public health policies.

Initially, we establish that HAT titres correlate with neutralising antibodies. We then use the HAT to investigate the antibody responses in 719 individuals consisting of home-dwelling older vaccinees (80–99-year-olds) and younger adults, in both those vaccinated with mRNA (BNT162b2) and in naturally infected individuals to the Wuhan-like virus. With VOC HAT we confirm that HAT titres can be used as a surrogate marker for neutralising antibody titres in vaccinated or infected individuals. Finally, we show that the HAT is readily adapted to finger prick testing.

## Methods
### Study participants
*Norwegian vaccine and infection cohorts.* A cohort of convalescents 415 infected individuals was prospectively recruited during the first Wuhan (pre-alpha) and delta pandemic waves in Bergen, Norway to compare the serological assays used in this study as described in[17–19]. For the comparison of vaccine and infection cohorts in Bergen Norway, we prospectively recruited two different age groups (home dwelling older and healthy younger adults) who received two doses of BNT162b2 mRNA COVID-19 vaccine at a 3-week interval during January 2021, and compared them to a group of 307 naturally infected individuals infected (1–89, median 47 years) with the Wuhan-like virus (D614G spike mutation) in February to April 2020[17,19] (Table 1). The older vaccinee group consisted of 96 home-dwelling elderly (80–99 years, median 86), 89 (92.7%) of whom were seronegative

and 7 had previous SARS-CoV-2 infection with detectable pre-vaccination antibodies. The younger adult group consisted of 316 vaccinees (23–77 years, median 38) of whom 309 adults had no history of confirmed SARS-CoV-2 PCR test. Four younger vaccinees were not vaccinated on day 21; they received their second vaccination at day 19 ($n = 1$), or day 23 ($n = 2$) or day 24 ($n = 1$). Seven younger individuals had previous SARS-CoV-2 infection and pre-existing antibodies. This study is compliant with all relevant ethical regulations for work with humans and conducted according to the principles of the Declaration of Helsinki (2008) and the International Conference on Harmonization (ICH) Good Clinical Practice (GCP) guidelines. All Bergen subjects provided written informed consent before inclusion in the study, which was approved by the Western Norway Ethics committee (#118664 and #218629, NIH ClinicalTrials.gov Identifier: NCT04706390). Demographics (gender, age), PCR test results and COVID-19-like symptoms were recorded in an electronic case report form (eCRF) in (REDCap® (Research Electronic Data Capture) (Vanderbilt University, Nashville, Tennessee). Clotted blood samples were collected on the day of vaccination, 3 weeks after receiving the first and 3–5 weeks (mean 55 days, standard deviation ± 5 days) after the second vaccine doses or 3–10 weeks after confirmed infection. Sera were separated and stored at −80 °C and heat-inactivated for one hour at 56 °C before use in the serological assays.

*UK convalescent cohort.* Informed signed consent was obtained from 420 blood donor in the NHS Blood and Transplant cohort for purposes of clinical audit, to assess and improve the services and the research, and specifically to improve knowledge of the donor population. The use of these anonymised donor samples to assess neutralising antibody levels using different assays was approved by the National Blood Supply Committee for Audit and Research Ethics of National Health Service Blood and Transplant Research and Audit Committee (BS-CARE; BSCR20047 and BSCR20051).

*Finger-prick and venous blood comparison.* For the comparison of finger-prick and venous blood, participants were recruited from Oxford University Hospitals NHS Foundation Trust when they were attending the research clinic with the Oxford Protective T Cell Immunology for COVID-19 (OPTIC) Clinical Team. Written informed consent was obtained from participants with different past infection and vaccination status. Seventy-eight paired finger-prick blood and venous blood in EDTA tubes were taken at the same time and analysed on the same day by the HAT assay. Human study protocols were approved by the research ethics committee at Yorkshire & The Humber-Sheffield (GI Biobank Study 16/YH/0247).

**Haemagglutination test (HAT).** The haemagglutination test (HAT)[15] was used to investigate the SARS-CoV-2 specific antibodies to the RBD of the ancestral virus (Wuhan-like, pre alpha) and to the VOC alpha (B.1.1.7), beta (B.1.351), gamma (P.1) and delta (B.1.617.2). Briefly, codon optimised IH4-RBD sequences of VOC containing amino acid changes in the RBDs B.1.1.7 (N501Y), B.1.351 (K417N, E484K, N501Y), P.1 (K417T, E484K, N501Y) and B.1.617.2 (L452R, T478K). IH4-RBD were expressed in Expi293F cells and purified by their c-terminal 6xHis tag using Ni-NTA chromatography.

The point HAT was performed in V-bottomed 96-well plate on the same day as the blood was collected. Whole blood was diluted 1 in 40 in Phosphate buffered saline (PBS) 50 μl of dilution was mixed with 50 μl 2 μg/ml IH4-RBD reagent in the test well. Anti-RBD monoclonal antibodies, EY-6A[20] or CR3022[21] (100 ng) were

**Table 1 The demographics of the old and healthy adult vaccinees and naturally infected subjects.**

| Characteristics | Norwegian (Bergen) cohort | | | | | UK (PHE) cohort |
| | Vaccinated | | Infected[&] | Wuhan convalescents[§] | Delta Convalescents[§] | Convalescents[§] |
| | Old (n = 96) | Adult (n = 316) | (n = 307) | (n = 378) | (n = 37) | (n = 420) |
|---|---|---|---|---|---|---|
| Age (median (age range)) | 86 (80–99) | 38 (23–77) | 47 (1–89) | 45 (1–89) | 17 (11–20) | 44 (19–65) |
| Sex (Female) | 61 (63%) | 214 (68%) | 159 (52%) | 216 (57%) | 21 (57%) | 114 (27%) |
| Comorbidity* | 81 (85%) | 41 (13%) | 136 (44%) | 154 (39%) | 2 (5%) | - |
| Immuno-suppression[#] | 14 (15%) | 4 (1%) | 12 (4%) | 11 (3%) | 0 (0) | - |

*Diabetes, chronic respiratory diseases, chronic heart diseases, neurological diseases, chronic kidney, or liver diseases, dementia, rheumatologic diseases, active cancer.
[#]Inherent immunosuppressive disease, HIV, organ transplant, chemotherapy, other immunosuppressive treatment/drugs.
[§]In correlation analysis, Fig. 1.
[&]In haemagglutination test (HAT) analysis, Fig. 2.
- Information was not available.

positive controls and negative controls were whole blood dilution mixed with PBS. All sera were pre-screened at a dilution of 1:40 in PBS in 96 well V well plates. If HAT positive, serum was double diluted in duplicate from 1:40 in 50 μl PBS giving final dilutions of 1:40 to 1:40,960. Equal volumes of human O negative red blood cells (~1% v/v in PBS)[15] and 2.5 μg/ml IH4-RBD of Wuhan-like or VOC (B.1.1.7, B.1.351, P.1 or B.1.617.2) (125 ng/well) were pre-mixed and 50 μl added per well. Negative controls (PBS) and positive controls (monoclonal antibodies CR3022 and EY-6A) were included in each run. Plates were incubated to allow red blood cells to settle for 1 hr and were read by tilting the plate for 30 s and photographing. Positive wells agglutinated and the HAT titre is defined as the last well in which the teardrop did not form. Partial teardrops were scored as negative.

The IH4-RBD reagents for each VOC were standardised by showing that agglutination of red cells occurred at the same endpoint dilution (~16 ng/well) of the well characterised human monoclonal antibody EY6A[15,20] for each VOC at a working dilution of IH4-RBD of 2 ug/ml (100 ng/well in 50 ul). All the RBDs of the VOC share the conserved class IV epitope recognised by EY6A.

**Enzyme-linked immunosorbent assay (ELISA).** SARS-CoV-2 antibodies were detected using the ELISA in Bergen, Norway as previously described, but with minor modifications (Supplementary Fig. 1)[13,17,18]. Sera were screened for IgG antibodies against the Wuhan RBD of the SARS-CoV-2 spike protein at a 1:100 dilution and all samples were run in duplicates. The sera were diluted in 1% milk, 0.1% Tween-20 solution in PBS and incubated for 2 h at room temperature in 96 well plates (Maxisorp, Nunc, Roskilde, Denmark) coated with 100 ng/well of the RBD antigen. Plates were washed with PBS containing 0.05% Tween (PBST) between each step. Bound IgG antibodies were detected with a horseradish peroxidase (HRP)-labelled secondary antibody (cat. no.: 2040-05, Southern Biotech, Birmingham, AL, USA) and the addition of the chromogenic substrate 3,3′,5,5′-tetramethylbenzidine (TMB; BD Biosciences, San Jose, CA, USA). Optical density (OD) was measured at 450/620 nm using the Synergy H1 Hybrid Multi-Mode Reader with the Gen5 2.00 (version 2.00.18) software (BioTek Instruments Inc., Winooski, VT, USA).

RBD positive sera were run in an additional ELISA, where the ELISA plates were coated with SARS CoV-2 spike protein (Wuhan, 100 ng/well). Sera were serially diluted in duplicate in a 5-fold dilution, starting from a 1:100 dilution, and the ELISA plates were incubated with diluted serum for 2 h at room temperature. Bound IgG antibodies were detected and measured as described for the RBD screening ELISA.

Positive controls were serum from a hospitalized COVID-19 patient with the pre-alpha virus and CR3022[22], whereas pooled

pre-pandemic sera (n = 128) were used as a negative control[18]. The mean endpoint titre was calculated for each sample. Samples with no detectable antibodies were assigned a titre of 50 for calculation purposes.

**Pseudotype-based neutralisation assay.** The pseudotype-based neutralisation assay was performed in biosafety level 2 laboratory in Bergen, Norway. The SARS-CoV-2 pseudotype virus was generated by co-transfection lentiviral vectors pHR'CMV-Luc, pCMVRΔ8.2, and pCMV3 construct encoding the Wuhan or delta spike protein into HEK293T cells as previously described[23]. The protease TMPRSS2 and human ACE2 encoding constructs were transfected into HEK293T to make target cells for the neutralisation assay. The lentiviral vectors and TMPRSS2-encoding constructs were a kind gift from Dr. Paul Zhou, Institute Pasteur of Shanghai, China. The ACE2-encoding construct was a kind gift from Dr. Nigel Temperton, University of Kent, UK. The SARS-CoV-2 Wuhan and delta spike-encoding constructs were purchased from Sino Biological. Serum samples were heat inactivated at 56 °C for 60 min, analysed in serial dilutions (duplicated, starting from 1:10). The SARS-CoV-2 pseudotype viruses corresponding to 20,000 to 200,000 relative luciferase activity (RLA) were mixed with diluted sera in 96-well plates and incubated at 37 °C for 60 min. Afterwards, ACE2-TMPRSS2 co-transfected HEK293T cells were added into 96-well plates and cultured for 72 h. RLA was measured by a BrightGlo Luciferase assay according to the manufacturer's instructions (Promega, Madison, WI, USA). The pseudotype-based neutralization (PN) titres ($IC_{50}$ and $IC_{80}$) were determined as the reciprocal of the sera dilution giving 50% and 80% reduction of RLA, respectively. Negative titres (<10) were assigned a value of 5 for calculation purposes.

**Virus strains.** The Wuhan-like strain used in the microneutralisation and virus neutralisation assays in Bergen Norway was the clinical isolate; SARS-CoV-2/Human/NOR/Bergen1/2020 (GISAID accession ID EPI_ISL_541970) and at Public Health England, UK the isolate England/02/2020[24] (GISAID accession ID EPI_ISL_407073). At Oxford, UK[25] the Wuhan-like strain was Victoria/01/2020 (GenBank MT007544.1, B hCoV-19_Australia_VIC01_2020_ EPI_ ISL_ 406844_ 2020-01-25, and alpha (B.1.1.72) virus was the H204820430, 2/UK/VUI/1/2020, the beta (B.1.351) (20I/501.V2.HV001) isolate and delta (B.1.617.2) (sequence identical to virus Genbank ID OK622683.1).

**Microneutralisation assay.** The microneutralisation (MN) assay was performed on 345 Bergen convalescent sera in a certified Biosafety Level 3 Laboratory in Norway[17–19] against a clinically isolated virus: SARS-CoV-2/Human/NOR/Bergen1/2020. Briefly, serum samples were heat inactivated at 56 °C for 60 min, analysed

in serial dilutions (duplicate, starting from 1:20), and mixed with 100 50% Tissue culture infectious doses (TCID$_{50}$) viruses in 96-well plates and incubated for 1 h at 37 °C. Serum-virus mixtures were transferred to 96-well plates seeded with Vero cells. The plates were incubated at 37 °C for 24 h. Cells were fixed and permeabilized with methanol and 0.6% H$_2$O$_2$, and incubated with rabbit monoclonal IgG against SARS-CoV2 NP (Sino Biological). Cells were further incubated with biotinylated goat anti-rabbit IgG (H+L) and horseradish peroxidase (HRP)-streptavidin (Southern Biotech). The reactions were developed with o-Phenylenediamine dihydrochloride (OPD) (Sigma–Aldrich). The MN titre was determined as the reciprocal of the serum dilution giving 50% inhibition of virus infectivity. Negative titres (<20) were assigned a value of 5 for calculation purposes.

The MN assay for the 420 convalescent UK samples was conducted in a certified Biosafety Level 3 as previously described at Public Health England (PHE), UK[24]. using the virus England/02/2020. Sera were heat inactivated at 56 °C for 60 min, before analyses in duplicate serial dilutions (starting from 1:20), and mixed with 100 TCID$_{50}$ viruses in 96-well plates and incubated for 1 h at 37 °C. Then, the cell suspension was added to the virus/antibody mixture[24] and incubated at 37 °C for 22 h. Cells were fixed and permeabilized before staining for NP antibodies, then biotinylated goat anti-rabbit IgG, followed by Extravidin-peroxidase. The reaction was developed with OPD and MN titres calculated, as described above.

At Oxford, UK the detection of antibodies to the Wuhan-like and VOC (alpha, B.1.1.7 and beta, B.1.351) used the method described in[25]. Briefly, quadruplicate serial dilutions of serum were preincubated with appropriate SARS-CoV-2 for 30 min at room temperature, then Vero CCL81 cells were added and incubated at 37 °C, 5% CO$_2$ for 2 h. A carboxymethyl cellulose-containing overlay (1.5%) was added, monolayers were fixed and stained for the nucleocapsid (N) antigen or spike (S) antigen using EY2A and EY6A monoclonal antibodies, respectively. After development the number of infectious foci were counted by ELISpot reader. Data were analysed using four-parameter logistic regression (Hill equation) in GraphPad Prism 8.3.

**Virus neutralisation assay.** The virus neutralisation (VN) assay was performed in a certified Biosafety Level 3 facility in Bergen, Norway[18]. Serum samples were tested against a clinically isolated virus: SARS-CoV-2/Human/NOR/Bergen1/2020 as previously described[18]. Briefly, serum samples were heat inactivated at 56 °C for 60 min, analysed in serial dilutions (duplicated, starting from 1:20), and mixed with 100 TCID$_{50}$ viruses in 96-well plates and incubated for 1 h at 37 °C. Mixtures were transferred to 96-well plates seeded with Vero cells. The plates were incubated at 37 °C

for 4–5 days, all wells were examined under microscope for cytopathic effect (CPE). The VN titre was determined as the reciprocal of the highest serum dilution giving no CPE. Negative titres (<20) were assigned a value of 5 for calculation purposes.

The delta virus neutralisation assay was performed at the University of Oxford, UK as previously described[26]. Briefly, serial two-fold serum dilutions from 1:20 were incubated with 50 TCID$_{50}$ virus in 96-well plates for 1 h before addition of 20,000 Vero E6 TMPRSS2 cells per well. Plates were incubated for 3 days before staining with amido black and CPE read by eye. Negative titres (<20) were assigned a value of 5 for calculation purposes.

**Statistics and reproducibility.** The two-tailed Mann–Whitney $U$ test with 95% confidence level was used to compare ranks in HAT titres between the older and adult vaccinees. The non-parametric two-tailed Spearman R correlation with 95% confidence interval was used to investigate the correlation between the antibody titres from different serological assays. All analyses were conducted in GraphPad Prism version 9.20.

## Results

Previous studies have shown that the HAT titre correlates with SARS-CoV-2 RBD binding and ACE2 blocking antibodies[15,27,28], and identified high titre (>100) neutralising sera with a sensitivity of 76.5%[28]. First, we used the World Health Organisation (WHO) approved human SARS-CoV-2 standards panel to confirm the relationship between spike specific binding and neutralising assay to the HAT assay (Table 2). Second, we investigated the relationship between endpoint HAT titres and neutralising antibodies using three neutralisation assays in convalescent sera from SARS-CoV-2 infected individuals from the first pandemic wave (pre-alpha) and the ongoing delta-wave in Bergen, Norway[17–19]. We then confirmed the results in an independent UK cohort[24].

**Correlation of neutralising antibodies and HAT titres.** In the Bergen Wuhan convalescents cohort, microneutralization 50% inhibitory concentrations (IC$_{50}$) titres were significantly associated with HAT titres (Spearman's R = 0.82, $p < 0.0001$) (Fig. 1b, Table 1, Supplementary Fig. 2). A HAT titre ≥ 40 detected 99% of samples with MN IC$_{50}$ ≥ 20, with positive predictive value (PPV) of 94%. A HAT titre > 480 predicted MN titres > 100 with a sensitivity 77% and PPV of 78%.

We extended these results by comparisons to a pseudotype neutralisation (PN) assay, and a classical live virus neutralisation (VN) assay with complete inhibition of Cytopathic Effect (100% CPE) as its endpoint (Fig. 1a, d, e)[18]. The correlation of HAT and PN titres were significant ($p < 0.0001$) at 50% (R = 0.79) and 80%

---

**Table 2 Comparison of the haemagglutination test antibody endpoint titres to the neutralisation and binding antibodies in the WHO anti-SARS-CoV-2 international standards.**

| Antibody test | WHO antibody standards* | | | |
| --- | --- | --- | --- | --- |
| | High | Mid | Low S, high N | Low S |
| HAT | 5120 | 640 | 5 | 5 |
| MN titre (IC$_{50}$) | 2298 | 240 | 55 | 21 |
| Neutralisation antibodies (IU/mL)# | 1473 | 210 | 58 | 44 |
| Anti-Receptor Binding Domain (RBD) (BAU/mL) | 817 | 205 | 66 | 45 |
| IgG anti-S1 (BAU/mL)# | 766 | 246 | 50 | 46 |
| IgG anti-Spike IgG (BAU/mL)# | 832 | 241 | 83 | 53 |
| Anti-N IgG (BAU/mL)# | 713 | 295 | 146 | 12 |

*The WHO anti-SARS-CoV-2 international standards (20/268 NIBSC, UK) contained high (20/150), mid (20/148), low spike (S) and high nucleocapsid (N) (20/144) and low S (20/140) human antibodies. The haemagglutination test (HAT) endpoint titres to the Wuhan-like virus were compared with microneutralisation (MN) titres and #neutralisation titres (IU/ml), and antibody binding (BAU/mL) as reported by NIBSC. Negative HAT tests are given a value of 5 for consistency.

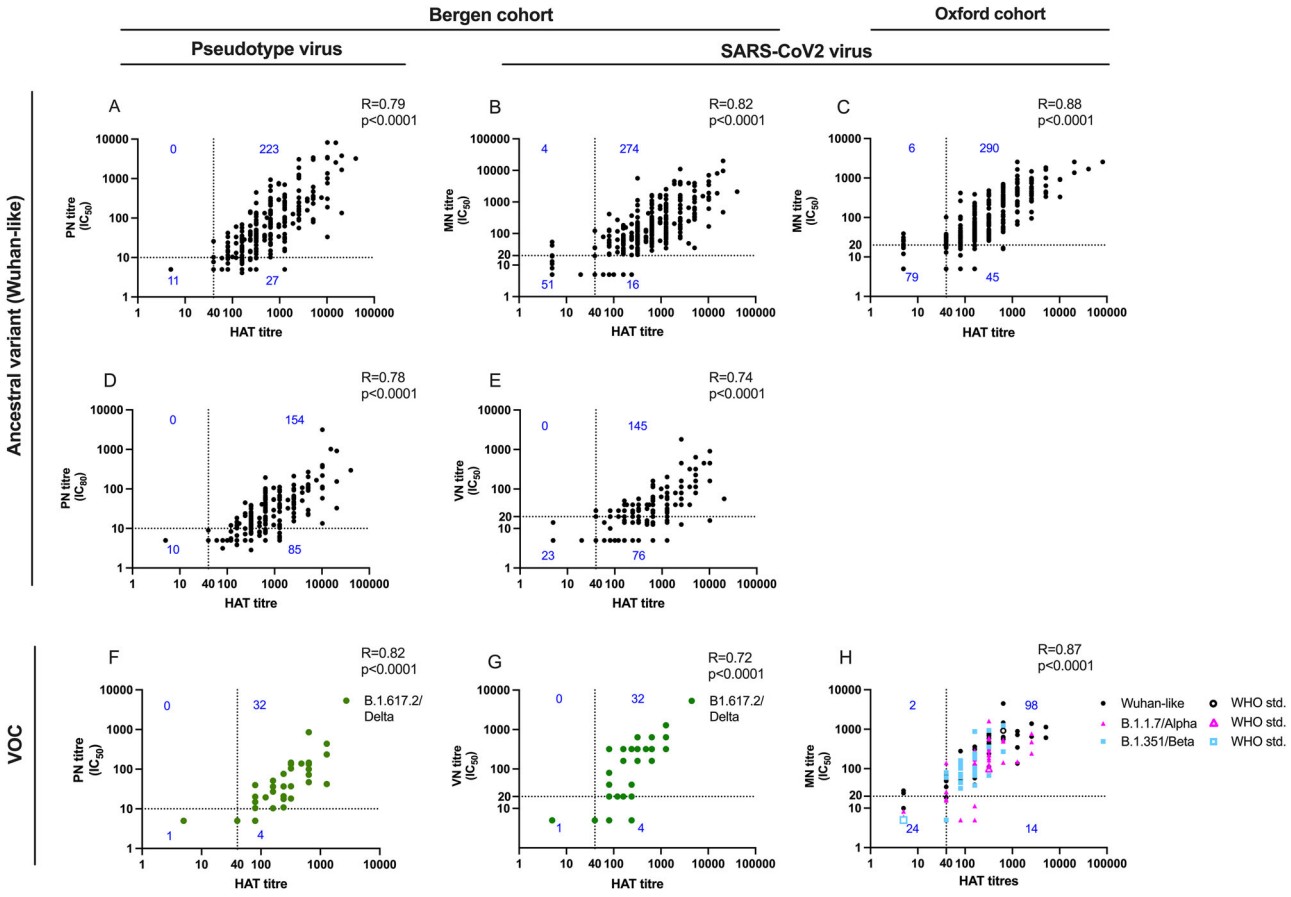

**Fig. 1 The correlation between haemagglutination test and SARS-CoV-2 neutralising antibodies.** Correlation of endpoint haemagglutination test (HAT) titres with neutralising antibody titres. In a cohort of infected individuals from the first (Wuhan-like) and delta pandemic waves (diagnosis by PCR from nasopharyngeal swabs or serology in Bergen, Norway) (**a–b**, **d–g**), Wuhan-like neutralising antibodies were measured using the pseudotype (PN) neutralisation assay at 50% ($PN_{50}$) (**a**) (R = 0.79, 95% confidence interval (CI): 0.73–0.83) and 80% ($PN_{80}$) (**d**) (R = 0.78, 95% CI: 0.72–0.82) inhibition of pseudotype virus infectivity, the microneutralisation (MN) 50% inhibitory concentration ($IC_{50}$) (**b**) (R = 0.82, 95% CI: 0.78–0.85) and virus neutralization (VN) 100% inhibition of cytopathic effect (**e**) (R = 0.74, 95% CI: 0.68–0.80) assays. Delta-like neutralising antibodies were measured in the PN assay at $PN_{50}$ (**f**) (R = 0.82, 95% CI: 0.67–0.90) and VN 50% inhibitory concentration (**g**) (R = 0.72, 95% CI: 0.51–0.85). Convalescent sera from 420 infected individuals in UK for whom neutralising antibody and HAT titre were measured (**c**) (R = 0.88, 95% CI: 0.86–0.90). The correlation between the HAT and 50% inhibition of neutralising antibody titres for Wuhan-like, and B.1.1.7 and B.1.351 VOC antibody titres performed at Oxford, UK (**h**) (R = 0.87, 95% CI: 0.82–0.90). HAT titres were measured in a set of donors either infected or vaccinated with one or two doses of the Pfizer BNT162b2 mRNA vaccine who had neutralising antibody levels to the ancestral Wuhan, B.1.1.7, B.1.351 or B1.617.2 live viruses. Open symbols represent the positive anti SARS-CoV-2 WHO standard (20/130). The Spearman R correlations and significant values are shown. In the MN assay, virus infectivity was measured by detecting the amount of nucleoprotein and also spike after 22–24 h incubation in Vero cells. In Bergen (**b**, **e**) the Wuhan-like local D614G virus hCoV-19/Norway/Bergen-01/2020 (GISAID accession ID EPI_ISL_541970) was used in a certified Biosafety Level 3 Laboratory. The dotted lines show the lowest detectable titre in each assay, all negative values were assigned the number 5 for consistency, and the sample size can be derived from adding the blue numbers in the quadrants together.

(R = 0.78) IC (Fig. 1a, d). Confirming our previous results, the VN titres correlated with HAT titres (R = 0.74, $p < 0.0001$) (Fig. 1e). HAT titres ≥ 40 detected 100% of samples with VN titres ≥ 20, but the PPV fell to 54% consistent with the classical VN assay having the more rigorous endpoint.

**Confirmation of correlation between neutralisation and HAT titres.** As interlaboratory variation has been reported for neutralisation assays, we confirmed the significant correlation between HAT and MN titres (R = 0.88, $p < 0.0001$) in an independent UK collection of 420 convalescent samples (Fig. 1c, Table 1, Supplementary Fig. 2). In close agreement with the Bergen cohort, a positive HAT ≥ 40 detected 98% of samples with MN $IC_{50}$ titre ≥ 20, with PPV of 87%. Similarly, for identification of high titre sera a HAT > 480 identified 75% of sera with MN $IC_{50}$ > 100 with PPV 86%. In summary, the HAT titres highly correlated with neutralisation titres in two independent

laboratories showing the utility of HAT as a rapid and inexpensive surrogate for the neutralisation test.

**Evaluation of HAT antibody responses in older and younger vaccinees.** Older adults have carried the burden of COVID-19 throughout the pandemic with increased risk of hospitalizations and death, and are prioritised for vaccination, although most vaccine licensure trials have excluded the oldest (>85 years old)[29]. As proof of principle, we used the HAT to investigate the Wuhan-like antibody responses in seronegative healthy younger adults (n = 309, median 37 years) and older home-dwelling adults (n = 89, 80–99 years, median 86 years) after the BNT162b2 mRNA COVID vaccine and in individuals naturally infected with the Wuhan-like strain (n = 307, median 47 years)[19] (Table 1, Fig. 2). A HAT titre of ≥40 was used as a cut-off to assess the propor of vaccine responders to the Wuhan-like virus[15]. Only 31% of older subjects responded after the first vaccination

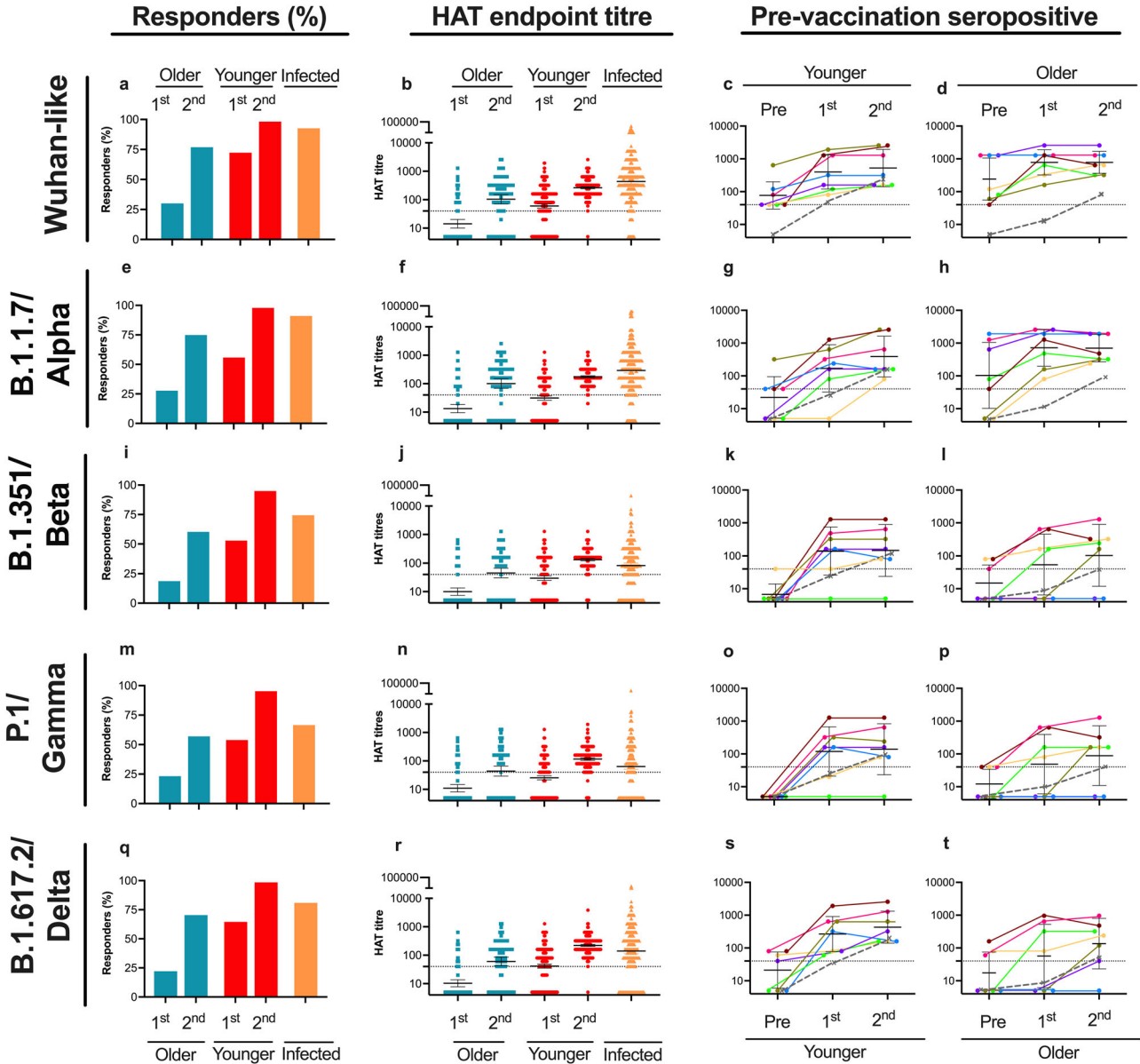

**Fig. 2 Haemagglutination test antibodies to the ancestral Wuhan SARS-CoV-2 virus and variants of concern in older and younger adult vaccinees and after natural infection.** Haemagglutination test (HAT) antibodies to the SARS-CoV-2 virus receptor binding domains of homologous founder virus (**a–d**: Wuhan-like) and variants of concern (**e–h** B.1.1.7 alpha; **i–l**: B.1.351 beta; **m–p**: P.1 gamma and **q–t**: B.1.617.2 delta). Endpoint HAT titres are presented in (**b–d**, **f–h**, **j–l**, **n–p**, **r–t**). The percentage of responders with haemagglutination test titre ≥ 40 (**a**, **e**, **i**, **m**, **q**) and endpoint HAT titres (**b**, **f**, **j**, **n**, **r**) in seronegative older ($n = 89$) and seronegative adults ($n = 309$) post 1st dose (3 weeks) and post 2nd dose (6–8 weeks after 1st dose) mRNA BNT162b2 COVID-19 vaccination. In infected individuals, convalescent serum was collected 3–10 weeks after SARS-CoV-2 confirmed infection (infected, $n = 307$) with D614G virus during the first pandemic wave (**a–b**, **e–f**, **i–j**, **m–n**, **q–r**). HAT endpoint titres to Wuhan-like and VOC in previously infected older individuals ($n = 7$) and adults ($n = 7$) who were vaccinated are shown in different colours, with the grey dashed line showing comparison of the geometric mean HAT titres for the corresponding seronegative (not previously infected) old ($n = 89$) and adult ($n = 309$) vaccinees (**c–d**, **g–h**, **k–l**, **o–p**, **s–t**). For endpoint HAT titres (**b–d**, **f–h**, **j–l**, **n–p**, **r–t**), negative values were assigned a value of 5. The geometric mean titres (GMT) and error bars with 95% confidence intervals are shown in black and each symbol represents one subject (**b**, **f**, **j**, **n**, **r**).

compared to 74% of younger vaccinees (Fig. 2a, Table 3). After the second dose, 78% of the older vaccinees had HAT titre of ≥40 compared to 94% of infected individuals and 99% younger vaccinees. Older people also had a significantly lower magnitude of response than younger adult vaccinees after both the first and second vaccine doses, with the exception after second dose against alpha (Table 4). In summary, the older adults had a blunted response after one dose of mRNA vaccine and required the second dose to increase the magnitude of the response.

**Development of variant of concern reagents for HAT.** Variants of concern have amino acid changes in their spike protein, and importantly in their RBD which may allow escape from neutralising antibodies. The alpha variant rapidly became the dominant strain in early 2021[2] with beta and gamma dominating in some geographical areas, and was subsequently replaced by the highly transmissible delta variant in 2021[30]. We developed HAT reagents for the VOC as they arose and confirmed a strong correlation between the alpha ($R = 0.79$, $p < 0.0001$) and beta

**Table 3 Comparison of Wuhan-like and variant of concerns haemagglutination test responders after one and two doses of mRNA vaccine in younger and older adults, as well as in previously infected individuals with SARS CoV-2.**

| Virus[#] | Responders n/N (%) | | | | |
|---|---|---|---|---|---|
| | Vaccinated | | | | Infected[3] |
| | Older[1] | Younger[2] | Older[1] | Younger[2] | |
| | Post 1st dose | | Post 2nd dose | | |
| Wuhan-like | 28/89 (31) | 228/309 (74) | 70/89 (78) | 308/309 (100) | 289/307 (94) |
| Alpha, B.1.1.7 | 26/89 (29) | 177/309 (57) | 68/89 (76) | 307/309 (99) | 284/307 (92) |
| Beta, B.1.351 | 18/89 (20) | 168/309 (54) | 55/89 (62) | 298/309 (96) | 233/307 (76) |
| Gamma, P.1 | 22/89 (25) | 171/309 (55) | 52/89 (58) | 299/309 (97) | 209/307 (68) |
| Delta B.1.617.2 | 21/89 (24) | 204/309 (66) | 64/89 (72) | 309/309 (100) | 253/307 (82) |

[1]89 seronegative older vaccinees 3 weeks after 1st and 3–5 weeks after 2nd dose of mRNA vaccine.
[2]309 younger adult vaccinees 3 weeks after 1st and 3–5 weeks after 2nd dose of mRNA vaccine.
[3]307 SARS-CoV-2 infected individuals with convalescent sera collected 4–6 weeks after infection.
[#]The viruses tested are the ancestral virus (Wuhan-like) and variant of concern (B.1.1.7 alpha; B.1.351 beta; P.1 gamma and B.1.617.2 delta) viruses. The data is presented as the subjects with HAT titre over 40 against the different variants, n/N and as percentage (%) of the whole group.

**Table 4 The haemagglutination test (HAT) antibody response to the Wuhan-like virus and variants of concern after one and two doses of mRNA vaccine and after SARS-CoV-2 infection in seronegative younger and older adults.**

| Virus[#] | Vaccinated | | | | | | | | | | Infected[3] |
|---|---|---|---|---|---|---|---|---|---|---|---|
| | Older[1] | | Younger[2] | | Older vs. younger | Older[1] | | Younger[2] | | Older vs. younger | |
| | Post 1st dose | | | | | Post 2nd dose | | | | | |
| | GM* | Fold-change$ | GM | Fold-change | P value[#] | GM | Fold-change | GM | Fold-change | P value[#] | GM |
| Wuhan-like | 14 | 2.8 | 60 | 11.9 | <0.0001 | 104 | 7.3 | 262 | 4.4 | <0.0001 | 438 |
| Alpha | 13 | 2.7 | 31 | 6.2 | <0.0001 | 101 | 7.6 | 175 | 5.6 | 0.3323 | 292 |
| Beta | 10 | 2.0 | 30 | 5.9 | <0.0001 | 45 | 4.5 | 133 | 4.5 | <0.0001 | 82 |
| Gamma | 11 | 2.2 | 25 | 5.1 | <0.0001 | 44 | 4.0 | 116 | 4.6 | 0.0036 | 64 |
| Delta | 10 | 2.0 | 41 | 8.3 | <0.0001 | 59 | 5.7 | 223 | 5.4 | <0.0001 | 141 |

[1]89 seronegative older vaccinees 3 weeks after 1st and 3–5 weeks after 2nd dose of mRNA vaccine.
[2]309 younger adult vaccinees 3 weeks after 1st and 3–5 weeks after 2nd dose of mRNA vaccine.
[3]307 SARS-CoV-2 infected individuals with convalescent sera collected 4–6 weeks after infection.
[#]The viruses tested are the ancestral virus (Wuhan-like) and variants of concern (B.1.1.7 alpha; B.1.351 beta; P.1 gamma; and B.1.617.2 delta) viruses.
*The data is presented as the geometric mean (GM) of the HAT titres. Negative values were assigned a value of 5 for calculation purposes.
$The fold change is shown in the vaccinated individuals from pre to post 1st dose and from post 1st to post 2nd vaccine dose. All individuals were seronegative (HAT < 40) at baseline.
[#]Two-tailed Mann–Whitney U test with 95% confidence level was used to compare ranks of HAT titres between the adults and the older vaccinees, with $P < 0.05$ considered significant. Statistically significant P values are in bold.

(R = 0.89, $p < 0.0001$) in a UK set of naturally infected and vaccinated donors[25] (Fig. 1h). As the delta variant dominates in Norway, we collected convalescent sera from from 37 infected individuals to confirm the relationship between HAT and pseudotype and virus neutralisation assays. A good correlation was observed for both pseudotype (R = 0.82, $p < 0.0001$) and virus neutralisation assays (R = 0.72, $p < 0.0001$) (Fig. 1f, g).

**HAT antibodies to variants of concern in older vaccinees**. We then investigated the breadth of the VOC response in vaccinees and infected subjects. Older vaccinees had the lowest number of responders and lower cross-reactivity after both one and two vaccinations. The second vaccination boosted the number of responders in older adults, from 20–29% to 58–76% to VOC (Fig. 2e, i, m, o). In older and younger vaccinees that responded to VOC, there was good cross-reactivity to alpha and delta, but less so to the beta and gamma in all groups after two doses of vaccine or infection. A similar but higher response pattern to different VOC was observed in infected individuals, with 92% to alpha, 82% to delta, 75% to beta and 68% to gamma compared to responses in 96–100% of younger adult vaccinees. In summary, two doses of mRNA vaccine or natural infection induced higher responses in younger adults to VOCs than in older vaccinees.

**Vaccine response in previously infected younger and older individuals**. Natural infection induces higher titres of SARS-CoV-2 specific antibodies in older individuals than in younger adults[17]. Previously infected older subjects ($n = 7$, median age 87 years), none of whom had been hospitalised, had higher pre-vaccination HAT titres to the Wuhan-like virus than previously infected younger adults ($n = 7$, median age 38 years). Previously infected older and younger adults developed high Wuhan-like and alpha cross-reactive antibody titres after one vaccine dose (Fig. 2c, d), although lower responses to other VOC. Cross-reactive titres were boosted in some of the older and healthy vaccinees after the second vaccination (Fig. 2g, h, k, l, o, p, s, t). In summary, previously infected older individuals develop high antibody titres after one vaccine dose comparable to healthy younger adults, which contrasts with the suboptimal antibody responses in SARS-CoV-2 naïve older vaccinees.

**The use of HAT as a point of care fingerprick test**. For the HAT to be implemented at low biosafety level and in resource limited settings, a fingerprick test using autologous patient erythrocytes could be used to rapidly identify populations with low titres to Wuhan-like and VOC SARS CoV-2 viruses. As advised by the Infectious Diseases Society of America guideline on serological

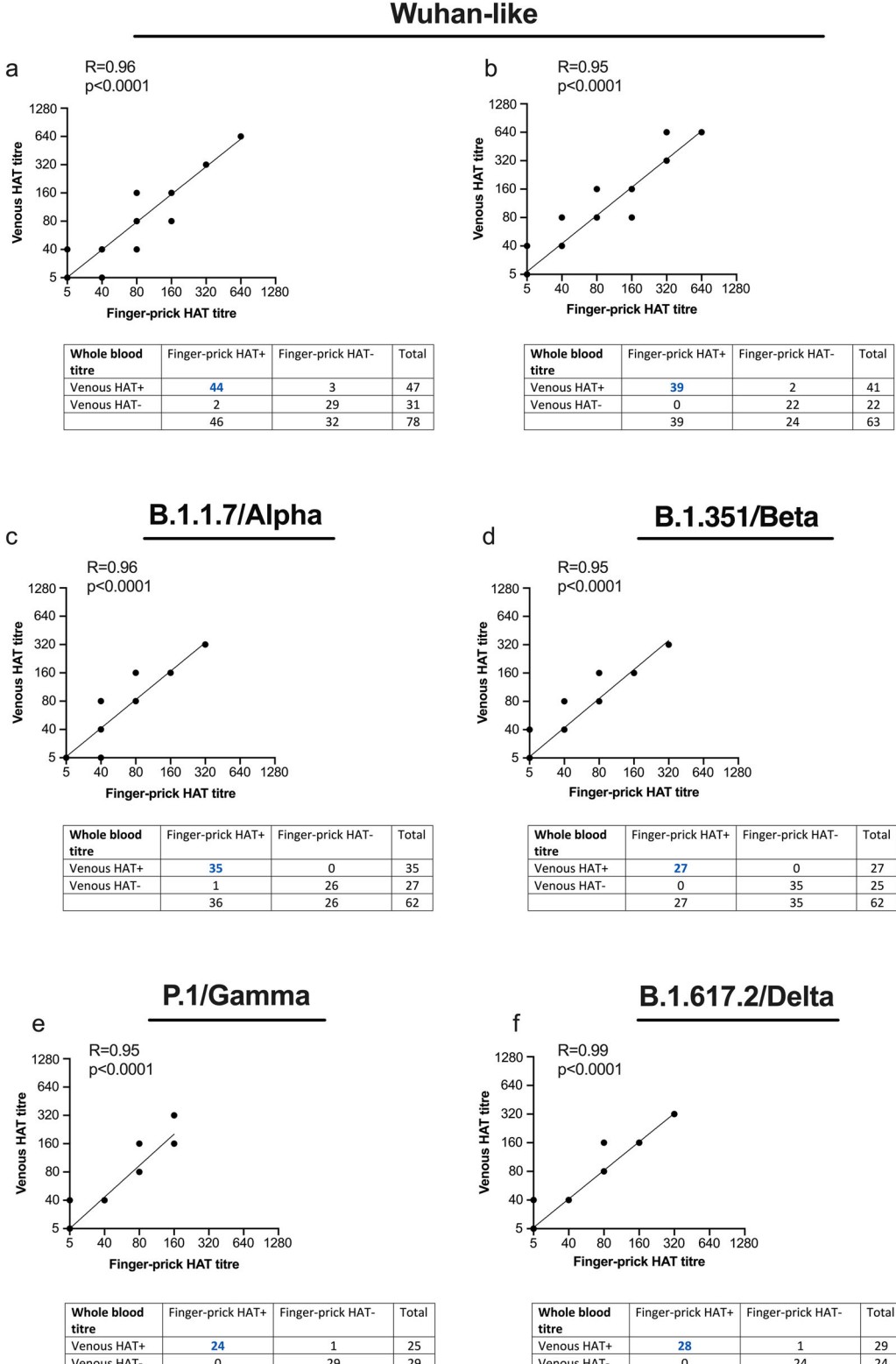

**Fig. 3 Correlation of finger-prick and venous blood samples using the haemagglutination test on Wuhan and variants of concern.** Correlation of paired finger-prick and venous blood samples collected from vaccinated healthcare workers. HAT titres are shown by a symbol that can represent on or more blood samples. Correlations were analysed by linear regression shown in graphs and tables. **a** The point haemagglutination test (HAT) showing the correlation between finger-prick and venous whole blood samples ($n = 78$). Haemagglutination was scored as shown in the contingency table. "HAT+" samples with haemagglutination and "HAT−" refers to no haemagglutination or endpoint titres <5. **b-f** Endpoint HAT titres of paired finger-prick and venous blood samples. Diluted finger-prick or venous whole blood samples (1 in 40 in phosphate buffered saline (PBS)) were centrifuged, and the supernatant was titrated in the HAT assay using IH4-RBD-reagents and autologous red blood cells (RBC) (washed and diluted 1 in 40 in PBS). **b** Wuhan-like ($n = 63$), **c** B1.1.7/Alpha ($n = 62$), **d** B.1.351/Beta ($n = 62$), **e** P1/Gamma, ($n = 54$), and **f** B.1.617.2/Delta ($n = 53$).

testing[31], we confirmed the comparability of fingerprick and venous blood for reactivity to all VOC (Fig. 3). The HAT can thus be rapidly adapted to test for antibodies to emerging VOC for large-scale screening of fingerprick blood samples with autologous erythrocytes.

## Discussion

The rapid evolution of SARS-CoV-2 VOC, particularly delta and omicron, with increased transmissibility and the possibility of escape from vaccine induced immunity, represents a considerable threat. There is a need for a low-cost rapid serological assay which can be used for large-scale screening globally without requiring specialised laboratory equipment to rapidly identify populations susceptibility to VOC. HAT is simple to perform, requires no special equipment, and can be done at the point-of-care in virtually any setting using a fingerprick sample. The HAT IH4-RBD VOC reagents are freely available for research[15]. Inter-laboratory comparability can be guaranteed by including HAT titrations on WHO approved standard sera (as shown in Table 2).

We demonstrate the versatility of the HAT in analysing susceptibility to VOC in home-dwelling older vaccinees showing the importance of two vaccine doses to achieve good cross-reactive antibody titres in older adults who have not been previously infected. Older and high-risk individuals were prioritised in the very first rounds of vaccination early in 2021 in Europe and America. Depending upon the decay in antibody titres over time, the HAT could be used to rapidly identify individuals who may need a booster vaccine dose to mount efficient antibody responses to VOC. We found the oldest age group had a decreased breadth of cross-reactive antibodies to VOC after the first vaccine dose, particularly to the beta and gamma viruses, in agreement with escape from neutralising antibodies[10]. Although vaccination induced cross-reactive antibodies against delta, milder break through infection with this variant in vaccinated subjects is becoming an increasing problem[11]. Reports of very high viral load during delta infections[32,33] may necessitate higher antibody titres to provide sterilising immunity and prevent infection.

In previously infected adults, only one dose of vaccine seems to be required to produce high levels of cross-reactive antibodies against the VOC[16,27]. Extending these findings, we found that in previously SARS-CoV-2 infected older adults, only one dose of vaccine was required to mount strong anamnestic responses, similar to younger vaccinees[34–40].

Caveats to our study are that most convalescent blood samples from our naturally infected cohort were collected during the first SARS-CoV-2 Wuhan-like wave, but we did include a small subset (n = 37) of delta infected individuals. Strengths are that we have confirmed the relationship between HAT and several neutralisation assays in two large cohorts in independent laboratories, showing that the relationship holds for VOCs, and included 719 individuals either infected and/or vaccinated, aged up to 99 years old. To our knowledge, this is the first study reporting antibody cross-reactivity to four VOC in this older age group.

Neutralising titres of between 10 and 30 in humans[1], depending upon the assay, have been reported to predict 50% protection from symptomatic infection, and much lower levels to protect against severe infection. Although the absolute HAT titres correlating with protection are not yet known, we demonstrated that the HAT titres correlated with neutralisation titres, and thus provide a surrogate test for neutralising antibodies. We suggest that a positive HAT titre of 40–80, equivalent to 1:40 dilution of whole blood obtained by fingerprick, would correlate with neutralising titres 10–30, and would predict protection. A prospective study to test this predicted relationship between HAT titres and

protection is now warranted. The HAT may also aid in evaluating and licensing of new COVID vaccines.

We predict that the lower HAT titres against VOC will lead to a more rapid decline in protective efficacy against variants, thus requiring booster vaccinations. The emergence of the highly infectious and transmissible delta and more worrying omicron VOC which have caused breakthrough infections in vaccinees highlights the importance of real-time cross-reactivity studies. Monitoring of population susceptibility of both previously infected subjects and vaccinees to VOC with increased transmissibility through simple serological assays can guide public health policy.

## Data availability

Source data behind figures are available in the Supplementary Data file.

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

## Acknowledgements

We are grateful to Professor Florian Krammer, Department of Microbiology, Icahn School of Medicine, Mount Sinai, New York, for supplying the RBD and spike constructs. We also thank the Research Unit for Health Surveys (RUHS), University of Bergen for collecting the home-isolated patients sera. The Influenza Centre is supported by the Trond Mohn Stiftelse (TMS2020TMT05), the Ministry of Health and Care Services, Norway; Helse Vest (F-11628, F-12167, F-12621), the Norwegian Research Council Globvac (284930); the European Union (EU IMI115672, FLUCOP, H2020 874866 INCENTIVE, H2020 101037867 Vaccelerate); the Faculty of Medicine, University of Bergen, Norway; Nanomedicines Flunanoair (ERA-NETet EuroNanoMed2 i JTC2016); and EU IMI Inno4vac 101007799. RUHS/FHU receives support from Trond Mohn stiftelsen (TMS). Public Health England is acknowledged for their financial support towards this work. DR was funded by the Department of Health and Social Care (DHSC)/UKRI/NIHR COVID-19 Rapid Response Grant (COV19-RECPLA). We are grateful to NIHR (UKRIDHSC COVID-19 Rapid Response Rolling Call, Grant Reference Number COV19-RECPLAS) and DHSC (PITCH), Huo family foundation (OPTIC), UKRI-CIC, NIHR Biomedical Research Centre, Oxford for providing facilities for finger-prick and venous sampling, and the medical student volunteers who took part. We are grateful for generous donations from WBP, EGB and ANB to the Townsend-Jeantet Prize Charitable Trust (Reg Charity No 1011770) which enable free distribution of the HAT test for detection of antibodies to the VOCs. Enquiries to alain.townsend@imm.ox.ac.uk.

## Author contributions

R.J.C., N.L. and A.T. designed the study, and together with N.U.E. and J.X. analysed the results and wrote the paper. N.U.E., J.X. and S.L. conducted the HAT assays. F.Z. and N.U.E. ran the neutralisation assays in Bergen, Norway. H.S., S.L.L., L.H., M.S., L.H., A.M., K.G.I.M., E.F., J.S.O. and K.A.B. recruited and followed up 345 SARS CoV-2 positive individuals and 412 vaccinees in Bergen and ran the lab assays. T.K.T., P.R., and L.S. developed and standardised the VOC HAT. S.D. and A.J. recruited and collected fingerprick and venous blood samples. D.R., W.S.J., D.N. and A.C.H. recruited the infected subjects and vaccinees in Oxford and ran the neutralisation assays for the VOC. H.H. and M.Z. collected and tested 420 convalescent samples at Public Health England Colindale UK for neutralising antibodies. SØ recruited and vaccinated the older patients. All authors reviewed the manuscript and approved the final version for publication.

## Competing interests

The authors declare no competing interests.

## Additional information

[1]Influenza Centre, University of Bergen, Bergen, Norway. [2]MRC Human Immunology Unit, MRC Weatherall Institute, John Radcliffe Hospital, Oxford, UK. [3]Department of Clinical Science, University of Bergen, Bergen, Norway. [4]Department of Microbiology, Haukeland University Hospital, Bergen, Norway. [5]Department of Medicine, Haukeland University Hospital, Bergen, Norway. [6]Eidsvåg Family Practice, Bergen, Norway. [7]Broegelmann Research Laboratory, University of Bergen, Bergen, Norway. [8]Department of Safety, Chemistry and Biomedical Laboratory Sciences, Western Norway University of Applied Sciences, Bergen, Norway. [9]Oxford University Hospitals NHS Foundation Trust, Oxford, UK. [10]NIHR Oxford Biomedical Research Centre, University of Oxford, Oxford, UK. [11]Sir William Dunn School of Pathology, University of Oxford, South Parks Road, Oxford OX1 3RE, UK. [12]Microbiology Services, NHS Blood and Transplant, Colindale, UK. [13]Nuffield Department of Medicine, Peter Medawar Building for Pathogen Research, University of Oxford, Oxford, UK. [14]Clinical, Research and Development, NHS Blood and Transplant, Oxford, UK. [15]Virology Reference Department, National Infection Service, Public Health England, Colindale, UK. [16]National Advisory Unit for Tropical Infectious Diseases, Haukeland University Hospital, Bergen, Norway. [22]These authors contributed equally: Nina Urke Ertesvåg, Julie Xiao, Alain Townsend, Nina Langeland, Rebecca Jane Cox. [23]These authors jointly supervised this work: Alain Townsend, Nina Langeland, Rebecca Jane Cox. *Lists of authors and their affiliations appear at the end of the paper. ✉email: alain.townsend@imm.ox.ac.uk; Nina.Langeland@uib.no; rebecca.cox@uib.no

## PHE Virology group

Monika Patel[15] & Robin Gopal[15]

## Oxford collaborative group

Leiyan Wei[2], Javier Gilbert-Jaramillo[11], Michael L. Knight[11], Alun Vaughan-Jackson[11], Maeva Dupont[11], Abigail A. Lamikanra[14], Paul Klennerman[17], Eleanor Barnes[10], Alexandra Deeks[17], Sile Johnson[18], Donal Skelly[9] & Lizzie Stafford[9]

[17]Peter Medawar Building for Pathogen Research, South Parks Rd, Oxford, UK. [18]Oxford University Medical School, Medical Sciences Division, University of Oxford, Oxford, UK.

## Bergen COVID-19 Research Group

Camilla Tøndel[16,19], Kanika Kuwelker[1,5], Bjørn Blomberg[3,5,16], Geir Bredholt[1,3], Therese Bredholt Onyango[1], Juha Vahokoski[1,16], Amit Bansal[1], Mai Chi Trieu[1], Håkon Amdam[1], Per Espen Akselsen[19], Trude Duelien Skorge[20], Liv Heiberg Okkenhaug[20] & Dagrunn Waag Linchausen[21]

[19]Department of Research and Development, Haukeland University Hospital, Bergen, Norway. [20]Occupational Health, Haukeland University Hospital, Bergen, Norway. [21]Bergen Municipality Emergency Clinic, Bergen, Norway.

