## [Peer Review File · Communications Medicine]

Reviewers' comments:

Reviewer #1 (Remarks to the Author):

The authors evaluate a haemagglutination test (HAT) based on SARS-CoV-2 spike receptor binding domain (RBD) linked to erythrocytes to assay antibody to SARS-CoV-2 spike RBD. They show that this assay correlates well with virus neutralizing antibody titres. This is shown with two cohorts of convalescent and post-vaccine sera from Norway and UK, respectively. They demonstrate that the HA assay can be carried out on fingerprint blood and show good correlation between HA titres on serum from venous bloods and from finger-prick samples. This would be of great use in low resourced settings, because it does not even require an ELISA reader because the results can be read by eye.

They then go on to use this assay with RBD of virus variants to assess RBD antibody to SARS-CoV-2 variants of concern. The most important data is found in figure 1F where they demonstrate good correlation between antibody titres to virus wild type, Alpha and Beta variants when measured by microneutralization and HAT assays vs. a panel of convalescent sera following wild type infection. They include the WHO standards in this evaluation. This is the correlation that may have been strengthened with more sera tested against this panel of variant viruses, and ideally, also using convalescent sera to the corresponding virus variants. This does not appear to have been done. While obtaining convalescent sera from people infected with Beta and Gamma VOC may be difficult to obtain, convalescent sera from Delta infected individuals should have been relatively easy to obtain in the UK and would be desirable.

It would be useful to discuss if the HAT assay is any different to an ELISA assay using the same RBDs? It is accepted that the HA assay has advantages in low resourced settings over ELISA. It is also true that HAT can be readily titered to end point to give antibody titers whereas antibody titers with ELISA is a bit more difficult to obtain. But is there any mechanistic reason why HAT titers may be different or superior to spike RBD ELISA?

Another issue that needs discussion in regard to assessing antibody to virus variants is the fact that these variants do have mutations or deletions in the spike NTD, a region that also elicits neutralizing antibodies. Using only the RBD may fail to recognize the NTD component to neutralization?

Specific comments:

The figures and tables cited in the results sometimes do not correspond to the relevant figure or table. For example, see line 89 and line 109 where the relevant figure should be Figure 1 rather than Extended data figure 1). Again, line 125/6 Extended Figure 1 is not the relevant figure.

Line 211: Extended data Table 1 is cited for the first time in the Discussion. But has not been cited or mentioned in Results, which it should be. It is fact quite important information and should be noted in the results.

Reviewer #2 (Remarks to the Author):

The manuscript is well written, with high impact in predicting human immune response versus Sars-Cov2 variants of concerns. The method assessed could be also an additional tool to use on studies for vaccines evaluation and assessment. The statistical analysis is quite basic but enough and overall

easy to understand. The assays are well described and seems to be easy to reproduce.

My opinion is that the manuscript is appropriate for the "communications medicine" journal, and could be published as it, with no additional revisions.

Reviewer #3 (Remarks to the Author):

This study used the hemagglutination test (HAT) developed by Townsend et al. and further confirmed that HAT titers were well correlated with neutralizing titers. This test was also applied to the evaluation of different antibody titers in older and younger vaccinees. Lastly, the authors showed as a proof of concept that the HAT can be used as a point of care fingerprick test. Overall, this study is a very nice extension of the study by Townsend et al. and is of interest to the researchers who wanted to have a quicker way to measure SARS-CoV-2 antibody levels in convalescent sera or vaccinee sera. The manuscript is well written and there are only a few places need editing.

1. Line 62, B.1.617.1 is not the delta variant, instead it's Kappa.
2. Lines 134-136 "Older adults . . . were not included in vaccine licensure trials" need to be clarified, as many clinical trials included older adults, unless the older adults have a specific definition.

Dear Editor,

We thank the reviewers for their positive and useful comments on our manuscript entitled “A rapid antibody screening haemagglutination test for predicting immunity to SARS CoV-2 Variants of Concern” (ID COMMSMED-21-0462-T). We appreciate their time in reviewing the manuscript and believe their input have improved our manuscript. The comments are answered in a point-by-point response below.

Reviewer #1 (Remarks to the Author):

The authors evaluate a haemagglutination test (HAT) based on SARS-CoV-2 spike receptor binding domain (RBD) linked to erythrocytes to assay antibody to SARS-CoV-2 spike RBD. They show that this assay correlates well with virus neutralizing antibody titres. This is shown with two cohorts of convalescent and post-vaccine sera from Norway and UK, respectively. They demonstrate that the HA assay can be carried out on fingerprint blood and show good correlation between HA titres on serum from venous bloods and from finger-prick samples. This would be of great use in low resourced settings, because it does not even require an ELISA reader because the results can be read by eye.

They then go on to use this assay with RBD of virus variants to assess RBD antibody to SARS-CoV-2 variants of concern. The most important data is found in figure 1F where they demonstrate good correlation between antibody titres to virus wild type, Alpha and Beta variants when measured by microneutralization and HAT assays vs. a panel of convalescent sera following wild type infection. They include the WHO standards in this evaluation. This is the correlation that may have been strengthened with more sera tested against this panel of variant viruses, and ideally, also using convalescent sera to the corresponding virus variants. This does not appear to have been done. While obtaining convalescent sera from people infected with Beta and Gamma VOC may be difficult to obtain, convalescent sera from Delta infected individuals should have been relatively easy to obtain in the UK and would be desirable.

It would be useful to discuss if the HAT assay is any different to an ELISA assay using the same RBDs? It is accepted that the HA assay has advantages in low resourced settings over ELISA. It is also true that HAT can be readily titrated to end point to give antibody titers whereas antibody titers with ELISA is a bit more difficult to obtain. But is there any mechanistic reason why HAT titers may be different or superior to spike RBD ELISA? Another issue that needs discussion in regard to assessing antibody to virus variants is the fact that these variants do have mutations or deletions in the spike NTD, a region that also elicits neutralizing antibodies. Using only the RBD may fail to recognize the NTD component to neutralization?

Specific comments:

The figures and tables cited in the results sometimes do not correspond to the relevant figure or table. For example, see line 89 and line 109 where the relevant figure should be Figure 1 rather than Extended data figure 1). Again, line 125/6 Extended Figure 1 is not the relevant figure.

Line 211: Extended data Table 1 is cited for the first time in the Discussion. But has not been cited or mentioned in Results, which it should be. In is fact quite important information and should be noted in the results.

We thank you for your thorough comments.

The reviewer is correct in that the HAT assay is essentially a binding assay, but with the difference from ELISA being that the antibodies detected have to be able to crosslink labelled red cells. We have found that the majority of Barnes Class 1 and 2 monoclonal antibodies

(that bind either side of the head of the RBD domain, block ACE2 and are strongly neutralising) cross-link very efficiently in the HAT assay. Greaney et al (<https://www.nature.com/articles/s41467-021-24435-8>) have shown that class 2 neutralising antibodies tend to dominate in polyclonal sera, at least to the Wuhan strain. We therefore suggest that the HAT assay favours the detection of neutralising antibodies, but this is not absolute, as non-neutralising antibodies to class 4 epitopes definitely also cross-link.

The HAT is a binding assay that correlates with MN (as all the binding assays do), but does not measure neutralising antibodies directly. Its advantages are its simplicity, it is quantitative, and its low cost. In situations where non-neutralising antibodies to RBD might dominate (such as detecting antibodies to Beta RBD after alpha infection with or without vaccination) the correlation may well falter. It is still however a reliable test for establishing detection of a level of antibodies to the RBD - many of which will be protective.

We apologise for the incorrect numbering and we have corrected this in the text. We have also moved the Extended data Table 1 of the WHO standards into the main material, and renamed it as Table 1, and these results are referred to in the result section.

We have collected 37 convalescent sera from people infected with the delta variant in Bergen, Norway. These sera were run in the HAT assay, the pseudotype neutralisation assay and the virus neutralisation assay with live delta virus. The data has been included into figure 1 as panels F and G, and showed a strong correlation with Spearman's correlation, $R=0.72-0.82$, $p<0.0001$.

Reviewer #2 (Remarks to the Author):

The manuscript is well written, with high impact in predicting human immune response versus Sars-Cov2 variants of concerns. The method assessed could be also an additional tool to use on studies for vaccines evaluation and assessment. The statistical analysis is quite basic but enough and overall easy to understand. The assays are well described and seems to be easy to reproduce.

My opinion is that the manuscript is appropriate for the "communications medicine" journal, and could be published as it, with no additional revisions.

We thank you for your positive comments.

Reviewer #3 (Remarks to the Author):

This study used the hemagglutination test (HAT) developed by Townsend et al. and further confirmed that HAT titers were well correlated with neutralizing titers. This test was also applied to the evaluation of different antibody titers in older and younger vaccinees. Lastly, the authors showed as a proof of concept that the HAT can be used as a point of care fingerprick test. Overall, this study is a very nice extension of the study by Townsend et al. and is of interest to the researchers who wanted to have a quicker way to measure SARS-CoV-2 antibody levels in convalescent sera or vaccinee sera. The manuscript is well written and there are only a few places need editing.

1. Line 62, B.1.617.1 is not the delta variant, instead it's Kappa.

2. Lines 134-136 "Older adults . . . were not included in vaccine licensure trials" need to be clarified, as many clinical trials included older adults, unless the older adults have a specific definition.

We thank you for your comments. We agree with the two issues you have raised and have now corrected the definition of the delta variant as well as included a more precise definition of the older adult groups not adequately represented in clinical studies.

REVIEWERS' COMMENTS:

Reviewer #1 (Remarks to the Author):

The revision has address the issues raised in the reviewers comments.

Reviewer #3 (Remarks to the Author):

The revision looks good.